# Persistent advanced HIV disease in rural KwaZulu-Natal, South Africa: Trends, characteristics, and the urgent need for targeted interventions

**Marcel K. Kitenge**[1]*, **Geoffrey Fatti**[1], **Ingrid Eshun-Wilson**[1,2], **Peter S. Nyasulu**[1,3]

**1** Division of Epidemiology and Biostatistics, Department of Global Health, Faculty of Medicine and Health Sciences, Stellenbosch University, Cape Town, South Africa, **2** Division of Infectious Diseases, School of Medicine, Washington University in St. Louis, Missouri, United States of America, **3** Division of Epidemiology and Biostatistics, School of Public Health, Faculty of Health Sciences, University of the Witwatersrand, Johannesburg, South Africa

* marcel.kanyinda@gmail.com

## Abstract

### Background

Advanced HIV disease (AHD) remains a persistent public health challenge. Data regarding the burden, characteristics and predictors of AHD is scarce specifically for rural settings of sub-Saharan Africa. This study aimed to describe trends in annual CD4 count distribution and to characterise adult persons living with HIV (PLWH) on ART who have AHD in rural KwaZulu-Natal, South Africa.

### Methods

A retrospective cohort design of annual CD4 count distribution was conducted among antiretroviral therapy (ART) patients aged 18 years and older. We used routinely collected data from adults receiving ART in Eshowe and Mbongolwane areas in KwaZulu-Natal, South Africa, between January 1, 2008, and June 30, 2021. Fine-Greys competing risks regression with proportional sub-distribution hazard models was used to determine factors associated with time to CD4 recovery.

### Results

A total of 34,729 patients were included of which 68.5% were females. The median age of the study sample was 33.5 years (interquartile range [IQR] 27–41 years), and the median CD4 count was 277 cells/μL (IQR, 149–452 cells/μL). The proportion of patients entering care with AHD declined over time from 62% in 2008 to 20% in 2021. Across all periods, those entering care with AHD were more likely to be men when compared to women (Relative risk [RR] 1.49; 95% 1.33–1.67). In addition, the proportion of patients with AHD who were ART-experienced increased from 4% in 2008 to 63% in 2021. Among ART-experienced, men were more likely to present with AHD compared to women (RR 1.79;

**Data availability statement:** Data cannot be shared publicly because of Protection Information Act (POPIA) in South Africa, which came into effect on 1 July 2021. Data are available from the KwaZulu-Natal Department of Health's Provincial Health Research Ethics Committee (hrkm@kznhealth.gov.za) for researchers who meet the criteria for access to confidential data. The data underlying the results presented in the study are available from Health Research and Knowledge Management (http://www.kznhealth.gov.za/hrkm.htm)/Ethics Committee (elizabeth.lutge@kznhealth.gov.za) for researchers who meet the criteria for access to confidential data.

**Funding:** The author(s) received no specific funding for this work.

**Competing interests:** The authors have declared that no competing interests exist.

95% CI 1.52–2.11). Among those with AHD, the cumulative incidence of CD4 recovery to >350 cells/µL was 3.21 (95% CI 3.13–3.29) per 100 adult-years follow-up time.

## Conclusion

Over time fewer patients with AHD are entering care in KwaZulu-Natal, South Africa. However, the proportion of PLWH entering care with AHD remains consistently high, affecting 1 in 4 PLWH accessing HIV services. In addition, there is an increasing number of ART-experienced patients with AHD. Implementation of male-friendly services, combined with intensified adherence support and re-engagement initiatives should be considered to reduce mortality risk for PLWH in rural regions in South Africa.

## 1. Background

Over the past decade, the South African national HIV program has made significant strides in expanding antiretroviral therapy (ART) coverage. By 2021, it had reached 74% of all individuals infected with the human immunodeficiency virus (HIV), with 5,588,062 adults on ART [1]. This expansion was facilitated by a series of guideline changes [2,3] allowing earlier initiation of ART and the implementation of a national strategic plan for prevention, treatment, and surveillance [4]. Since September 1, 2016, South Africa has adopted the "Treat all policy," enabling persons living with HIV (PLWH) to commence treatment irrespective of CD4 count [5,6].

Several studies have demonstrated that initiating ART at higher CD4 counts is associated with near-normal life expectancy and significantly reduces the risk of onward HIV transmission [7–10]. Conversely, a CD4 count below 200 cells/µL or WHO stage 3 or 4 event (Advanced HIV Disease for those ART-naïve or returning to care after interruption in treatment) strongly predicts severe morbidity, comorbidities and mortality [11,12].

Despite efforts to promote early ART initiation, there is limited data on the burden of Advanced HIV Disease (AHD) among both ART-naïve and ART-experienced patients under routine programmatic conditions, particularly in HIV high-prevalence and low-income rural settings. Historically, rural areas have faced challenges in healthcare accessibility, especially for specialized care for PLWH [13]. Rural PLWH encounter greater barriers to HIV care than their urban counterparts [13–15]. Gaps remain in understanding the burden of AHD in rural South Africa and detailed information from rural areas remains scarce. Data on the burden of AHD and predictive its clinical and demographic characteristics is needed to guide equitable distribution of screening, prophylaxis, rapid ART initiation, and intensified adherence interventions to PLWH presenting with AHD. This means, making AHD care accessible and available not only in urban, high-functioning health services but also in hard-to-reach rural areas, to ensure none are left behind.

Quantifying the burden of AHD in rural settings will allow implementers to consider and target interventions to ultimately reduce the risk of HIV morbidity and mortality for all PLWH.

To address this evidence gap, we describe trends of annual CD4 count distribution among adult patients receiving ART in rural KwaZulu-Natal, South Africa. Furthermore, we characterize PLWH with CD4 < 200 cells/µL according to demographic characteristics, prior viral load, and ART experience, and assess time to CD4 recovery.

## 2. Methods

### Design

This was a retrospective cohort design of annual CD4 cell count data among adults receiving ART between January 1, 2008, and June 30, 2021, using routinely collected ART program data at 10 public health facilities.

### Setting and participants

South Africa implemented the Universal Test and Treat (UTT) on September 1, 2016. Under this strategy, all public health facilities are required to scale up ART initiation for all HIV-infected individuals according to UTT guidelines, with an emphasis on providing same-day initiation for individuals newly diagnosed with HIV who are clinically and psychologically ready for lifelong ART [16]. KwaZulu-Natal province in South Africa has 1.9 million PLWH, 1.1 millions of whom were receiving ART in 2023 [17]. Since 2011, Médecins Sans Frontières (MSF) and the Department of Health (DoH) of KwaZulu-Natal have been implementing a community-based HIV/TB project. The project involves 10 primary health facilities and two hospitals in the Eshowe and Mbongolwane areas. The areas are predominantly rural areas with one semi-urban market town, with a combined population of 114,490 in 2021 [18]. According to a population-based survey conducted by MSF, these areas exceeded the UNAIDS 90-90-90 targets by achieving 90-94-95 in 2018. The survey also revealed that HIV prevalence in the study areas was an estimated 26.4 among 15–59 years old adults [15]. Our study included all individuals aged 18 years and older living with HIV who were receiving ART and had undergone a baseline CD4 measurement when entering HIV care, along with subsequent CD4 cell measurements between January 2008 and June 2021.

### Data sources and variables

Data for this study were obtained from Tier.net, an electronic ART database developed by the University of Cape Town. Tier.net is used operationally in public health facilities in South Africa to monitor baseline clinical care and clients' outcomes over time [19]. For this analysis, the data on clients initiating ART in Eshowe and Mbongolwane areas, KwaZulu-Natal, was extracted from Tier.Net on 31st May 2022 by the District Health Information Systems Manager and handed over to the study authors. We utilised demographic and clinical data documented at ART initiation as well as during follow-up.

### Definitions and outcomes

The primary outcome was the percentage of patients with AHD among ART-naïve and ART-experienced patients. We defined AHD among ART-naïve patients as patients who were newly diagnosed and entered care with CD4 < 200 cells/μL. ART-experienced patients were those who entered care with CD4 count ≥ 200 cells/μL and who experienced a drop in CD4 count to < 200 cells/μL during follow-up (exposure to ART for at least 6 months or more), including those lost to follow-up and who returned to care. Time to CD4 recovery was defined as the time from the last CD4 < 200 cells/μL to the first CD4 > 350 cells/μL during the follow-up [20]. Lost to follow-up (LTFU) was defined as 90 days late for a visit by the South African ART programme, with the date of the last visit used as the date of loss to follow-up. We also defined prior attendance as unknown if the patients had neither a ≥ 90-day interruption in care nor regular attendance or continuously in care.

### Statistical analysis

Continuous variables were summarised using mean and standard deviation if normally distributed, otherwise median, and interquartile range (IQR) were used, and categorical variables

were summarised using percentages. We used mixed-effect logistic regression models assuming a random effect for each facility to assess the probability of having AHD among ART-naïve patients at ART start adjusted for age, sex, calendar year, and TB status.

Poisson regression models adjusted for clustering at the facility level were used to identify predictors of AHD among ART-experienced patients. Independent variables with $P < 0.1$ in univariable analysis were included in the multivariable model. We present risk ratios (RR) and odds ratios (OR) to show the relative strength of the association.

We fitted a competing risks (CR) model to estimate the cumulative incidence of CD4 recovery to >350 cells/µL following a CD4 < 200 cells/µL. A CR analysis assumes that patients were exposed to at least two competing events (CD4 recovery to > 350 cells/µL or death or LTFU). The event of interest is CD4 recovery to > 350 cells/µL, while the others are competing events that prevent the outcome of interest from occurring [21–23]. We used Fine-Gray competing risks regression with a proportional sub-distribution hazard (SHR) model adjusted for clustering at the facility to model cumulative incidence, quantify the instantaneous risk of failure from CD4 recovery, and assess the overall impact of covariates on the incidence of CD4 recovery to >350 cells/µL, treating death and LTFU as a competing risks [21,24]. Measures of associations were presented with 95% confidence intervals and factors with $p < 0.05$ were considered statistically significant. Data were analysed using STATA 17 (StataCorp College Station, TX).

### Ethical statement

The study was approved by the Health Research Ethics Committee (HREC) at Stellenbosch University (S21/11/253) and the KwaZulu-Natal Department of Health's Provincial Health Research Ethics Committee (NHRD Ref: KZ_202204_018). Since this study involved routinely collected data, informed consent from participants was not obtained. The named ethics committees waived the requirement for consent because the study was retrospective in nature and used anonymised individual data.

## 3. Results

### Demographic characteristics and trends of AHD among ART-naïve patients

Our analysis included 34,729 PLWH who initiated ART between January 2008 and June 30, 2021, with 35.5% initiating ART during the universal test and treat era era (Fig 1). Table 1 shows demographic characteristics of patients included in the analysis, 68.5% were female, median age was 33.5 years (IQR 27–41), median CD4 count was 277 cells/µL (IQR 149–452), and median follow-up was 41 months (IQR 12–82). CD4 counts rose over the years, peaking in 2016 before stabilizing, with men showing lower counts than women (S1 Fig). The proportion of patients with AHD declined from 62% in 2008 to 25% in 2019 but has remained constant among ART-naïve patients since 2017 (Fig 2).

### Factors associated with AHD among ART-naïve patients

In the mixed-effects logistic regression, being male(Adjusted odds ratio[aOR] 1.80; 95% CI 1.71 to 1.90) and aged 25–31, 32–38, or 39–45 years were associated with higher odds of AHD at ART initiation compared to ages 18–24. The aORs for these age groups were 1.65 (95% CI 1.52–1.79), 2.15 (95% CI 1.98–2.34), and 2.22 (95% CI 2.03–2.44), respectively. Additionally, having tuberculosis (aOR 2.46; 95% CI 2.19–2.76) was associated with higher odds of AHD at ART initiation (Table 2). Starting ART between 2012–2015 (aOR 0.36; 95% CI 0.34–0.38) or

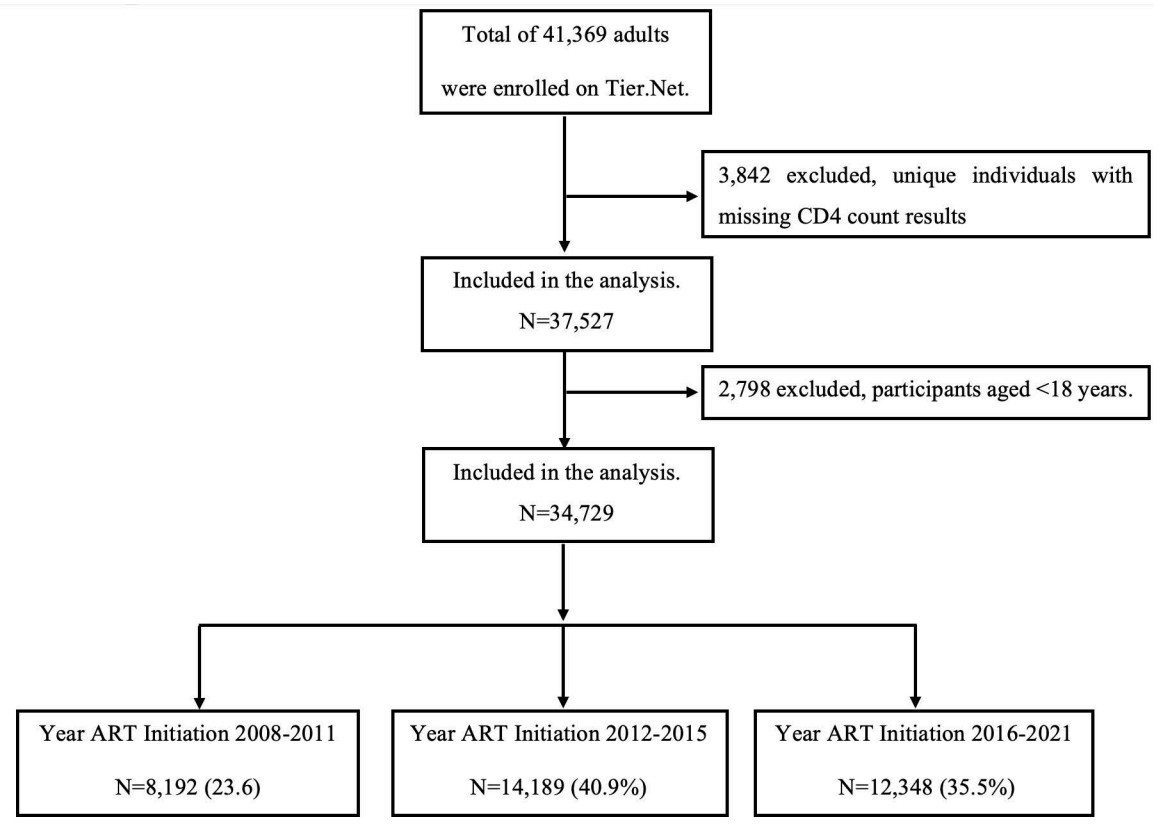

**Fig 1. Flowchart, overview of patients included in the analysis.**

2016-2021 (aOR 0.22; 95% CI 0.21–0.24) was associated with lower odds of AHD at initiation compared to 2008-2011. While the proportion of PLWH entering care with AHD declined over time, the proportion among females decreased notably, whereas it increased for males (Table 3). In an adjusted Poisson regression model, men had a higher rate ratio (RR) of entering care with a first CD4 count < 200 cells/μL than women, rising from RR 1.17 (95% CI 1.11-1.23) in 2010 to RR 1.65 (95% CI 1.54–1.76) in 2020 (Table 3). The excess AHD burden among men compared to women was evident across all age groups.

## Burden and trends of AHD among ART-experienced patients

During the study period, a total of 4,257 PLWH had CD4 < 200 cells/μL at some stage during ART (AHD among ART-experienced), of which 1,923 (45.2%) were men, 174 (4.2%) were on TB treatment and 2,868 (67.4%) were on TDF+FTC/3TC based regimen. During the study period, the proportion of AHD among ART-experienced increased from 4% in 2008 to 63% in 2021 (Fig 3). Forty-two percent of those ART-experienced with AHD had either had a prior episode of attendance unknown or interrupted treatment for 90 days or more (S2 Fig). Furthermore, among those with AHD on ART, 14% of patients had a prior high viral load (VL) ≥ 1000 copies/ml, 69% had a VL < 1000 copies/ml, and 16% had no VL results (S3 Fig).

## Factors associated with AHD among ART-experienced patients

In the multivariable Poisson regression model assessing factors associated with AHD among ART-experienced patients adjusted for clustering at facility level and other

                                                                                  

**Table 1. Participants' demographic at ART initiation.**

| Variables | Total (N = 34,729) |
|---|---|
| **Age group in years, n (%)** | |
| 18–24 | 5,686 (16.4) |
| 25–31 | 10,076 (29.0) |
| 32–38 | 8,325 (24.0) |
| 39–45 | 5,141 (14.8) |
| >45 | 5,501 (15.8) |
| **Sex n (%)** | |
| Female | 23,790 (68.5) |
| Male | 10,939 (31.5) |
| **Distribution of baseline CD4 counts, n (%)** | |
| 0–99 | 5,453 (15.7) |
| 100–199 | 7,245 (20.9) |
| 200–349 | 9,026 (26.0) |
| ≥350 | 13,005 (37.4) |
| **On TB Treatment at ART initiation, n (%)** | |
| No | 34,035 (98.0) |
| Yes | 694 (2.0) |
| **WHO Stage, n (%)** | |
| 1 | 12,056 (34.7) |
| 2 | 6,259 (18.0) |
| 3 | 5,900 (17.0) |
| 4 | 798 (2.3) |
| Missing | 9,716 (28.0) |
| **Year of ART initiation, n (%)** | |
| 2008–2011 | 8,192 (23.6) |
| 2012–2015 | 14,189 (40.9) |
| 2016–2021 | 12,348 (35.5) |
| **ART regimen on enrollment n (%)** | |
| ABC + 3TC | 153 (0.4) |
| AZT + 3TC | 336 (1.0) |
| TDF+FTC/3TC | 28,986 (83.5) |
| Stavidine + 3TC/FTC | 4688 (13.5) |
| Missing | 556 (1.6) |
| **Time on ART n (%)** | |
| < 1 years | 8,588 (24.8) |
| 1-3 years | 7,770 (22.2) |
| > 3 years | 18,379 (53.0) |

ART: antiretroviral therapy, ABC: Abacavir, AZT, Zidovudine, 3TC: Lamivudine, TDF: Tenofovir, FTC: Emtricitabine

baseline characteristics, men were more likely to present with CD4 count < 200 cells/μL during ART (RR 1.63; 95% CI 1.46–1.82) compared to women. The effect of gender was strongest among men aged 39–45 years (RR 2.95; 95% CI 1.97–4.43) compared to women. Moreover, PLWH in the age groups of 25–31 years, 32–38 years, and 39–45 years exhibited a higher risk of presenting with CD4 counts <200 cells/μL at some point during their treatment when compared to those aged 18–24 years. The relative risks (RR) for these

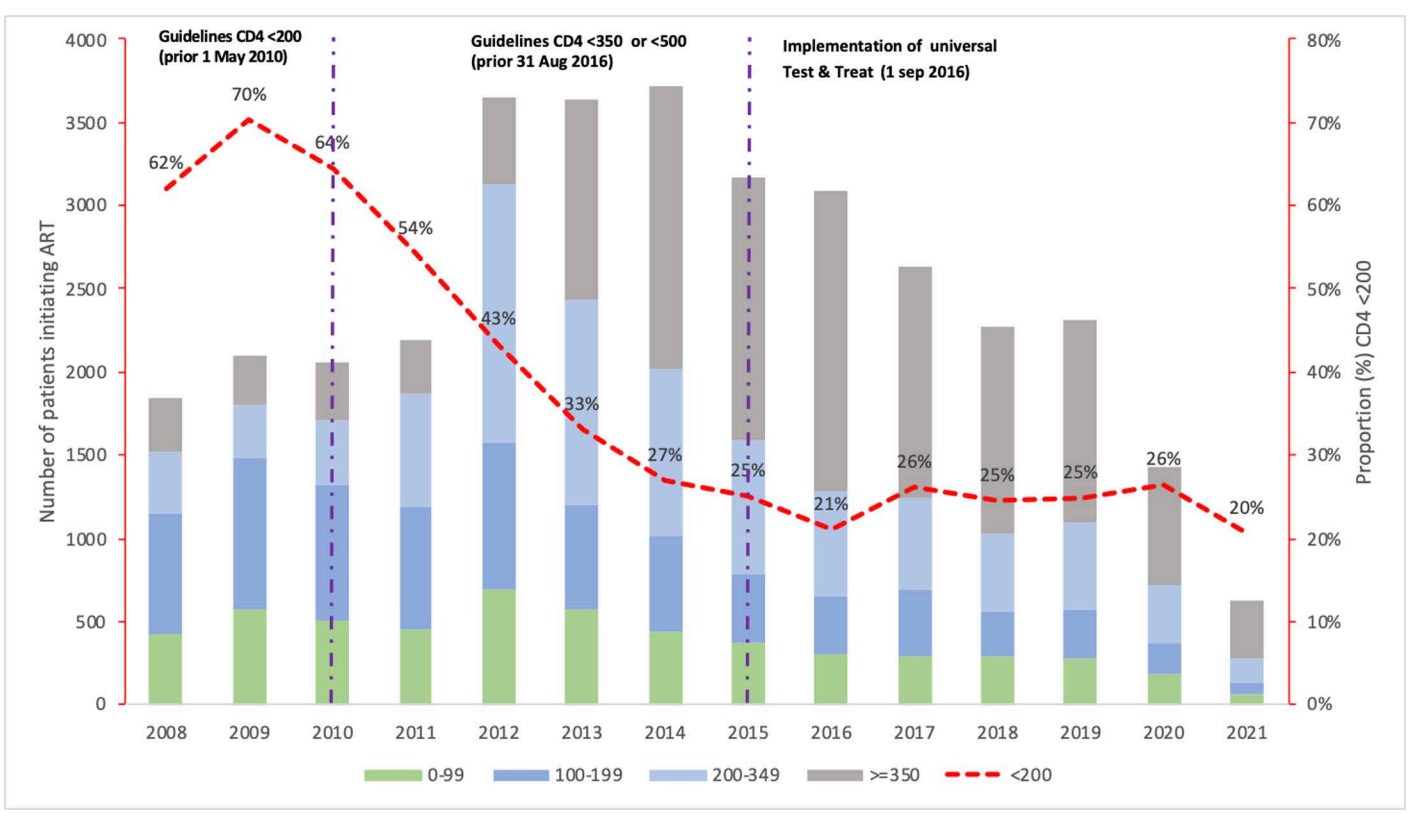

**Fig 2. Number and percentage of patients initiating by CD4 count.**

age groups were 1.35 (95% CI 1.18–1.55), 1.61 (95% CI 1.26–2.10), and 1.74 (95% CI 1.37–2.22), respectively. Later year of ART initiation was associated with a reduced risk of AHD during ART. However, TB treatment was not associated with AHD during ART (RR 1.10; 95% CI 0.89 to 1.36) ([Fig 4]).

### Time to CD4 recovery to >350 cells/μL following a CD4 <200 cells/μL among ART-naïve and ART-experienced patients

A total of 5,769 PLWH had documented an increase in CD4 counts to > 350 cells/μL (or CD4 recovery to > 350 cells/μL) at a median of 24.3 months (IQR; 12.6–41.5) from the date of CD4 <200 cells/μL. The cumulative incidence of CD4 recovery to > 350 cells/μL was 3.21 cases (95% CI 3.13–3.29) per 100 adult-years. At 6, 12, and 24 months, the incidence rates were 1.26 (95% CI 1.14–1.38), 1.98 (95% CI 1.88–2.10), and 2.61 (95% CI 2.51–2.71) cases per 100 adult-years, respectively. CD4 recovery was higher in females than males, with incidence declining with increasing age ([S4] and [S5 Figs]).

In the multivariable competing risks regression model for time to CD4 recovery > 350 cells/μL following a CD4 < 200 cells/μL, adjusted for health facility clustering ([Table 4]), CD4 recovery was higher among women than men (aSHR 1.38; 95% CI 1.31–1.46). CD4 recovery was also higher in those aged 25–31 years (aSHR 1.66; 95% CI 1.48–1.85) compared to ages 18–24, with increasing age was associated with an increasing relative incidence of CD4 recovery to >350 cells/μL. Among TB/HIV co-infected, those on TB treatment showed a better CD4 recovery (aSHR 1.19; 95% CI 1.03–1.38) than those not on TB treatment. Starting

**Table 2. Factors associated with AHD among ART-naive patients at initiation of ART.**

| | With AHD (n = 12,695) n (%) | No AHD (n = 22,034) n (%) | Unadjusted OR (95% CI) | Adjusted OR (95% CI) |
|---|---|---|---|---|
| **Sex** | | | | |
| Female | 7,447 (58.7) | 16,343 (74.2) | Ref | Ref |
| Male | 5,248 (41.3) | 5,691 (25.8) | 1.89 (1.81–1.98) | 1.80 (1.71–1.90) |
| **Age Groups** | | | | |
| 18–24 years | 1,087 (8.6) | 4,599 (20.9) | Ref | Ref |
| 25–31 years | 3,291 (25.9) | 6,785 (30.8) | 1.93 (1.79–2.10) | 1.65 (1.52–1.79) |
| 32–38 years | 3,546 (27.9) | 4,779 (21.7) | 2.85 (2.63–3.10) | 2.15 (1.98–2.34) |
| 39–45 years | 2,375 (18.7) | 2,766 (12.5) | 3.24 (2.97–3.54) | 2.22(2.03–2.44) |
| >45 years | 2,396 (18.9) | 3,105 (14.1) | 2.87 (2.63–3.14) | 1.88 (1.71–2.06) |
| **On TB Treatment at ART start** | | | | |
| No | 11,911 (93.8) | 21,384 (97.1) | Ref | Ref |
| Yes | 784 (6.2) | 650 (2.9) | 1.91 (1.71-2.13) | 2.46 (2.19-2.76) |
| **Year of ART Initiation** | | | | |
| 2008-2011 | 5,134 (40.5) | 3,058 (13.9) | Ref | Ref |
| 2012-2015 | 4,585 (36.1) | 9,604 (43.6) | 0.36 (0.34-0.38) | 0.36 (0.34-0.38) |
| 2016-2021 | 2,976 (23.4) | 9,372 (42.5) | 0.25 (0.23-0.27) | 0.22 (0.21-0.24) |

Data are n (%). Percentages in the left-hand column were calculated with the total in the header as the denominator. All other percentages used the column as the denominator. ART = antiretroviral therapy; OR = Odds ratio. Results from mixed-effects logistic regression model showing the probability of AHD at ART start, adjusted for clustering at facility level.

**Table 3. Adult females, adult males, and All adults with first CD4 count test < 200 cells/μL, by sex and calendar year of test.**

| Females | | | | Males | | | Total | | | RR* (95% CI) |
|---|---|---|---|---|---|---|---|---|---|---|
| Year | <200 cells/μL n (%) | ≥200 cells/μL n (%) | Females with first CD4 Test, No. | <200 cells/μL n (%) | ≥200 cells/μL n (%) | Males with first CD4 Test, No. | <200 cells/μL n (%) | ≥200 cells/μL n (%) | Persons with first CD4 Test, No. | |
| 2008 | 785 (60) | 524 (40) | 1309 | 361 (67) | 179 (33) | 540 | 1146 (62) | 703 (38) | 1849 | Reference |
| 2009 | 996 (69) | 453 (31) | 1449 | 481 (74) | 169 (26) | 650 | 1477 (70) | 622 (30) | 2099 | 1.03 (0.97 to 1.10) |
| 2010 | 834 (60) | 547 (40) | 1381 | 485 (72) | 186 (28) | 671 | 1319 (64) | 733 (36) | 2052 | 1.17 (1.11 to 1.23) |
| 2011 | 751 (51) | 731 (49) | 1482 | 438 (62) | 272 (38) | 710 | 1189 (54) | 1003 (46) | 2192 | 1.17 (1.09 to 1.25) |
| 2012 | 901 (37) | 1510 (63) | 2411 | 681 (55) | 562 (45) | 1243 | 1582 (43) | 2072 (57) | 3654 | 1.37 (1.31 to 1.42) |
| 2013 | 683 (27) | 1864 (73) | 2547 | 522 (48) | 571 (52) | 1093 | 1205 (33) | 2435 (67) | 3640 | 1.38 (1.29 to 1.46) |
| 2014 | 560 (21) | 2124 (79) | 2684 | 444 (43) | 595 (57) | 1039 | 1004 (27) | 2719 (73) | 3723 | 1.40 (1.31 to 1.50) |
| 2015 | 432 (19) | 1815 (81) | 2247 | 362 (39) | 563 (61) | 925 | 794 (25) | 2378 (75) | 3172 | 1.45 (1.31 to 1.50) |
| 2016 | 347 (16) | 1838 (84) | 2185 | 303 (34) | 598 (66) | 901 | 650 (21) | 2436 (79) | 3086 | 1.49 (1.33 to 1.67) |
| 2017 | 347 (20) | 1406 (80) | 1753 | 342 (39) | 532 (61) | 874 | 689 (26) | 1938 (74) | 2627 | 1.57 (1.46 to 1.70) |
| 2018 | 289 (19) | 1220 (81) | 1509 | 274 (36) | 493 (64) | 767 | 563 (25) | 1713 (75) | 2276 | 1.55 (1.37 to 1.75) |
| 2019 | 264 (18) | 1206 (82) | 1470 | 308 (37) | 532 (63) | 840 | 572 (25) | 1738 (75) | 2310 | 1.72 (1.57 to 1.88) |
| 2020 | 181 (20) | 739 (80) | 920 | 195 (39) | 308 (61) | 503 | 376 (26) | 1047 (74) | 1423 | 1.65 (1.54 to 1.76) |
| 2021 | 73 (16) | 370 (84) | 443 | 55 (30) | 128 (70) | 183 | 128 (20) | 498 (80) | 626 | 1.36 (1.16 to 1.60) |
| **All years** | **2835 (31)** | **16347 (69)** | **23790** | **5251 (48)** | **5688 (52)** | **10939** | **12694 (37)** | **22035 (63)** | **34729** | |

Data are presented as No (%) unless otherwise indicated. Confidence intervals; RR=risk ratio. Rate ratio estimated via Poisson regression, adjusted for clustering at facility level, **95% CI**: *Males with First CD4 count 0-199 cells/mL compared to Females.

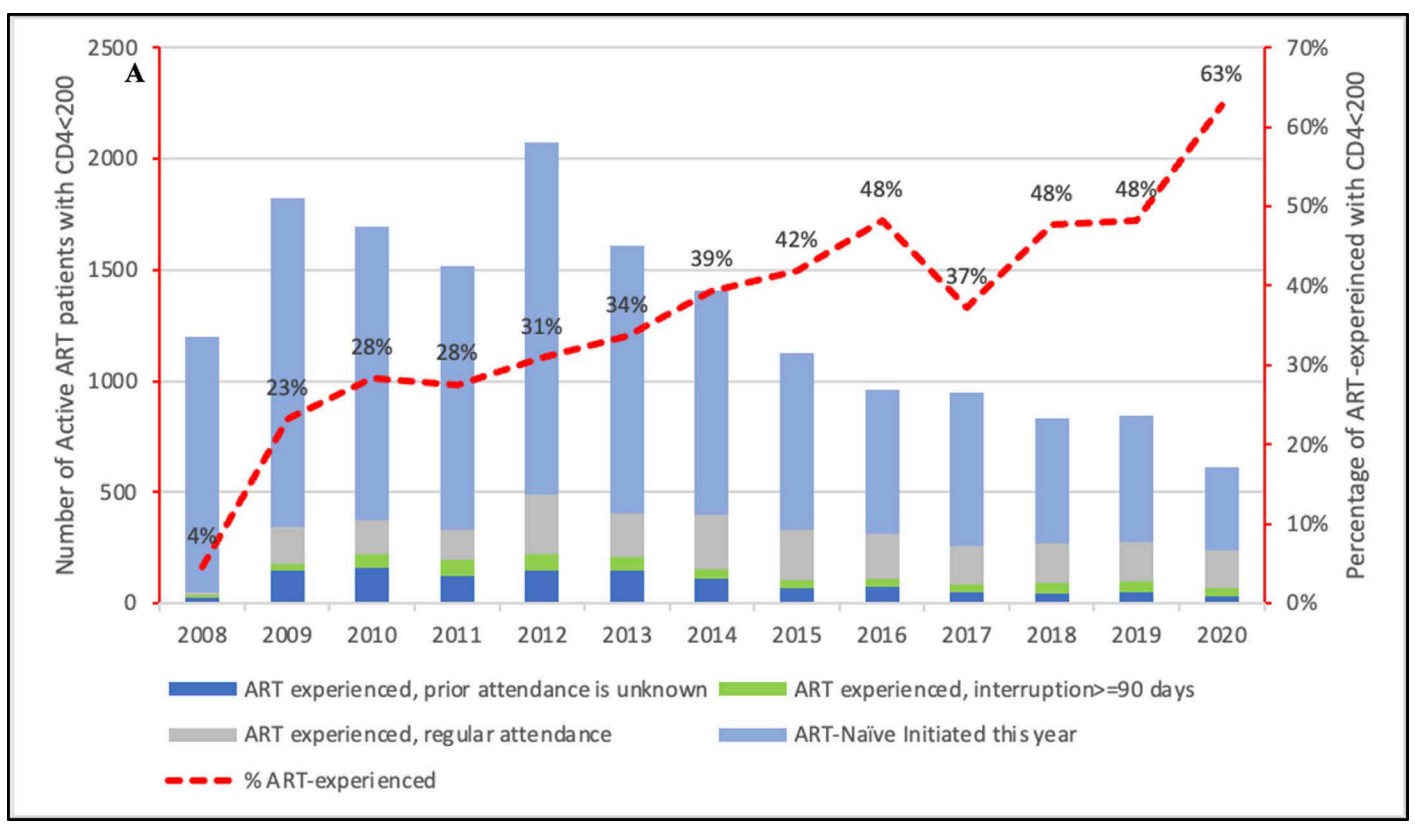

**Fig 3. Number and proportion of patients on ART with CD4 < 200 cells/ μL by previous ART-experience category.**

ART between 2012–2015 (aSHR 0.60; 95% CI 0.56–0.65) or 2016-2021 (aSHR 0.54; 95% CI 0.49–0.59) versus 2008–2011 was associated with a reduced incidence of CD4 recovery >350 cells/μL.

Finally, patients who were taking ABC + 3TC (aSHR 0.51; 95% CI 0.30–0.88), TDF+FTC/3TC (aSHR 0.75; 95% CI 0.30–0.88) and AZT + 3TC (aSHR 0.92; 95% CI 0.85–0.98) containing regimen were associated with a relative incidence of CD4 recovery to > 350 cells/μL compared to Stavudine + 3TC/FTC).

## 4. Discussion

Our findings indicate that although the proportion of PLWH entering care with AHD in rural KwaZulu-Natal, South Africa, has decreased over 12 years, it has remained consistently high (affecting 1 in 4 PLWH presenting to HIV services) despite the expansion of HIV testing services and the successful implementation of ART scale-up. This reduction in AHD is however offset by an increasing number of PLWH re-presenting to HIV services who have previously initiated ART and now return with AHD after a treatment gap. Our analysis also suggests that CD4 recovery to >350 cells/μL took longer than 12 months and the relative incidence of CD4 recovery to >350 cells/μL was higher among females compared to males, after adjustment for baseline covariates.

During the study period, we found that 36.6% of PLWH had a CD4 count <200 cells/μL at baseline, this is marginally higher than the global estimates [25] and estimates reported in other studies within SSA [26–28]. Notably, certain studies in SSA indicate even higher proportions of AHD at

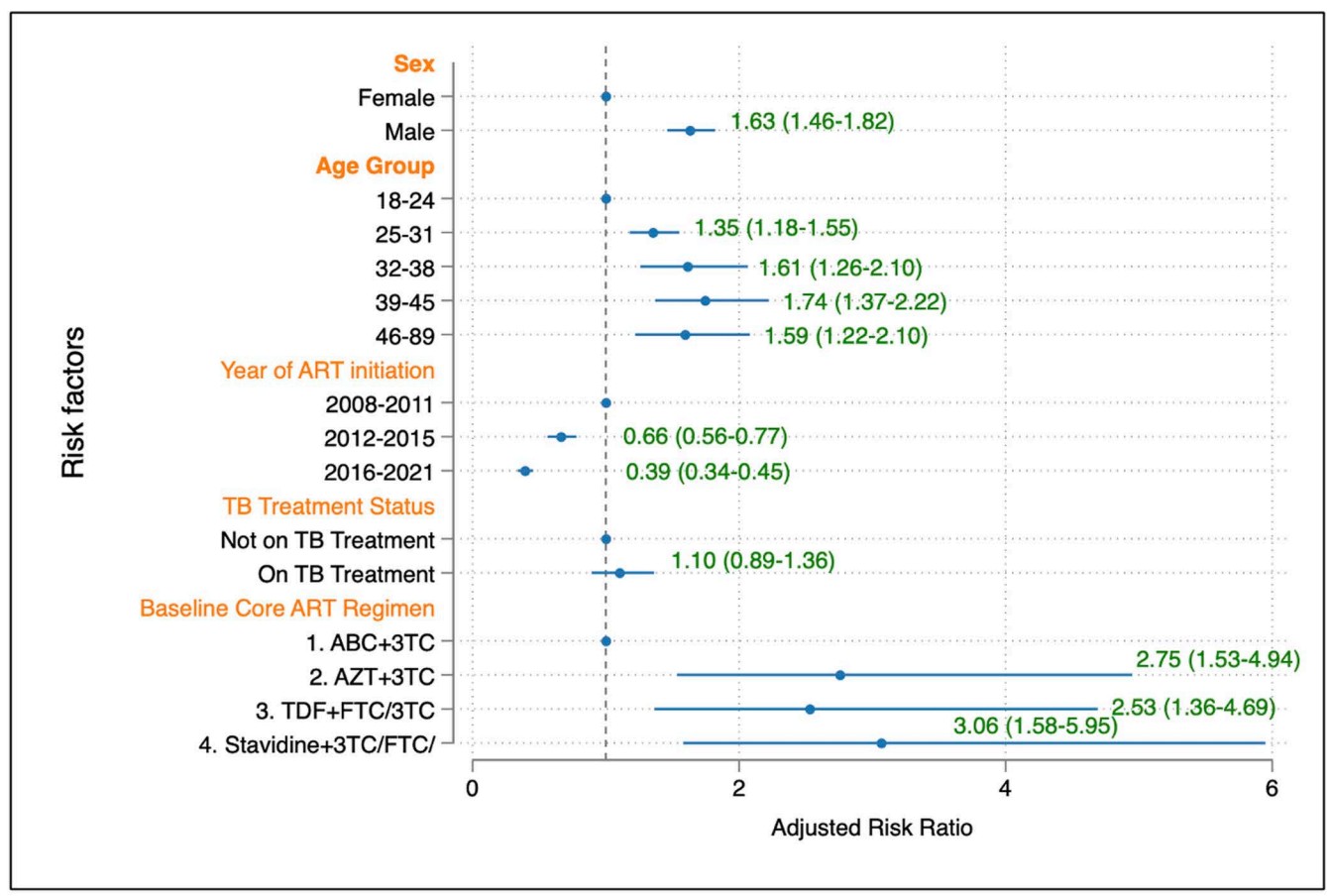

*ART: antiretroviral therapy; ABC: Abacavir , AZT: Zidovudine, 3TC: Lamivudine, TDF: Tenofovir, FTC: Emtricitabine, TB: Tuberculosis.*

**Fig 4. Factors associated with Advanced HIV Disease among ART-experienced patient.** *ART: antiretroviral therapy; ABC: Abacavir, AZT: Zidovudine, 3TC: Lamivudine, TDF: Tenofovir, FTC: Emtricitabine, TB: Tuberculosis.*

diagnosis, reaching up to 62% in rural Tanzania and Ethiopia [29–31]. A recent meta-analysis showed that 43.4% of PLWH in South Africa between 2010–2022 initiated ART with AHD [28].

The declining trends in AHD among ART-naïve patients in this study reflect shifts in ART guidelines. Initially, ART eligibility was restricted by low CD4 thresholds (<200 cells/μL until 2010 and <350 cells/μL until 2013), necessitating repeated CD4 testing to determine treatment eligibility [32–34]. However, with rising CD4 thresholds and the 2016 "treat-all policy" recommending immediate ART upon HIV diagnosis, fewer PLWH began treatment with AHD [2]. To further reduce the prevalence of AHD among ART-naïve patients, strategies should focus on early HIV testing and linkage to care, including primary healthcare referrals, adherence support post-hospital discharge, and implementing WHO's universal ART eligibility [35,36]. Enhancing community awareness of AHD is also essential, as awareness remains limited [36].

Noticeable sex disparities emerged and consistently increased over the study period, with men showing an increasing relative risk of presenting with AHD in later years. This trend is in keeping with findings from SSA, where men have consistently shown higher odds of

**Table 4. Factors associated with CD4 recovery to > 350 cells/µL following a CD4 < 200 cells/µL from a competing risks regression model among ART-naïve and ART-experienced patients.**

| | Univariable SHR (95%CI) | p-values | Multivariable SHR (95% CI) | p-values |
|---|---|---|---|---|
| **Sex** | | | | |
| Male | Ref | | Ref | |
| Female | 1.25 (1.19–1.32) | <0.001 | 1.38(1.31–1.46) | <0.001 |
| **Age Groups** | | | | |
| 18–24 years | Reference | | Reference: | |
| 25–31 years | 1.71 (1.53–1.91) | <0.001 | 1.66 (1.48–1.85) | <0.001 |
| 32–38 years | 2.26 (2.03–2.52) | <0.001 | 2.11 (1.89–2.35) | <0.001 |
| 39–45 years | 2.44 (2.19–2.73) | <0.001 | 2.24 (1.99–2.52) | <0.001 |
| >45 years | 2.28 (2.04–2.55) | <0.001 | 2.07 (1.84–2.32) | <0.001 |
| **On TB Treatment at ART start** | | | | |
| No | Reference | | Reference | |
| Yes | 0.92 (0.79–1.06) | 0.266 | 1.19 (1.03–1.38) | 0.016 |
| **Year of ART Initiation** | | | | |
| 2008–2011 | Reference | | Reference | |
| 2012–2015 | 0.52 (0.49–0.55) | <0.001 | 0.60 (0.56–0.65) | <0.001 |
| 2016–2021 | 0.45 (0.41–0.49) | <0.001 | 0.54 (0.49–0.59) | <0.001 |
| **ART regimen on enrolment** | | | | |
| Stavudine + 3TC/FTC | Reference | | Reference | |
| ABC + 3TC | 0.32 (0.19–0.55) | <0.001 | 0.51 (0.30–0.88) | 0.015 |
| AZT + 3TC | 0.64 (0.50–0.81) | <0.001 | 0.75 (0.30–0.88) | 0.023 |
| TDF+FTC/3TC | 0.57 (0.54–0.61) | <0.001 | 0.92 (0.85–0.98) | 0.018 |

SHR = Sub-distribution hazard ratio, ART: antiretroviral therapy, ABC: Abacavir, AZT: Zidovudine, 3TC: Lamivudine, TDF: Tenofovir, FTC: Emtricitabine, TB: Tuberculsis.

AHD at ART initiation [26,30], a disparity attributed to multiple factors. The rapid expansion of ART—primarily focused on women and children—benefited from strong political support [37] and funding [38], enabling earlier access to care for these groups [39]. In contrast, HIV-positive men face unique barriers, including suboptimal health-seeking behaviors and varying motivations for treatment compared to women [40,41]. ART programs should prioritize identifying and addressing these access barriers, including clinic-level adaptations to foster male-friendly environments and ensure equitable care access for both sexes [42].

Our study also found that there was an increase in the number of PLWH with AHD who were ART-experienced (PLWH who have previously initiated ART and are re-engaging with AHD after a period without effective ART or failed therapy) from 4.0% in 2008 to 62.0% in 2020. Of these AHD patients who were ART-experienced, 27% had either had a prior attendance unknown or interrupted treatment for 90 days or more and 14% had a detectable VL of ≥ 1000 copies/ml, suggesting either adherence problems or failed therapy. Additionally, among those ART-experienced patients with AHD with documented VL, 69.0% were virally suppressed. These findings align with results from a multicountry cohort study conducted in Kenya and Democratic Republic of Congo, which examined the characteristics and hospital outcomes of HIV-positive patients. The study revealed that 65.1% of patients were admitted with AHD. Among these inpatients, 71.7% had previous ART experience at admission, with a median duration of 55.9 months on ART [43]. Additionally, annual CD4 snapshots from a public sector cohort in Western Cape Province, South Africa, showed ART-experienced patients with AHD increased from 14.3% to 56.7% between 2008 and 2017 [44].

This demonstrates that in a high HIV-prevalence setting such as South Africa, a significant number of PLWH with AHD may not be identified if viral load testing is used alone. This finding contributes to the growing literature on patients who experience immunosuppression despite ART and virologic response. The subset of PLWH experiencing AHD despite prolonged ART and viral load suppression has been noted in several other studies from the region [43,45,46].

Until recently, AHD was seen mainly as a result of delayed presentation, leading to the belief that increased testing and earlier diagnosis could resolve the issue. While late presentation remains a factor, AHD is now mostly observed in individuals who disengaged from care, returning only upon falling ill. The impact of cycling in and out of care on AHD has only recently been recognized, with established health risks from treatment interruptions, including a rapid CD4 decline within two months [47,48]. These findings highlight the need for strategies to close HIV care gaps, promote re-engagement [49], and encourage early return to care before severe deterioration occurs [49,50].

Our analysis found that CD4 recovery to > 350 cells/μL took over 12 months and was slower in men than in women, aligning with findings from other studies in SSA and South Africa [51–53]. The cumulative incidence of CD4 recovery to > 350 cells/μL was initially rapid after ART initiation but plateaued around 12 months [54]. Additionally, CD4 recovery was higher in those on TB treatment, supporting previous evidence [55] that PLWH with TB achieve immune recovery comparable to other groups in South Africa. In contrast, younger individuals showed slower CD4 recovery to > 350 cells/μL [51].

## Strengths and limitations

This study represents one of the initial studies to quantify the prevalence of AHD among both ART-naïve and ART-experienced patients in rural areas characterized by high HIV prevalence and low-income settings under routine programmatic conditions, allowing for an estimation of the prevalence of AHD under programmatic conditions at public sector facilities. This approach maximises the generalizability of the findings.

Secondly, we used data from routine public sector clinics adhering to South Africa's Department of Health guidelines and used programmatic outcomes definitions. A key strength of this study is its large sample size and follow-up period of up to 7 years, enabling CD4 recovery to > 350 cells/μL descriptions beyond the usual 1–2 years. However, interpretation should be done with caution due to limitations inherent in observational study designs. Patients with only one CD4 measurement may have died or disengaged from care, potentially introducing selection bias and limiting the applicability of findings to the full population in the HIV program.

Thirdly, recommendations for CD4 count monitoring frequency shifted between 2010 and 2014. From 2010 to 2011, CD4 count assessments were conducted at ART initiation, 6 months, 12 months, and annually. Starting in 2012, CD4 measurements occurred at ART initiation, 12 months, and annually or as clinically needed. Consequently, fewer individuals in later years had both baseline and follow-up CD4 counts. Variability in adherence to CD4 and VL monitoring guidelines across facilities may impact cohort eligibility and follow-up CD4 count frequency, with significant variation observed in CD4 monitoring at 6 or 12 months across sub-districts and facilities [56].

## 5. Conclusion

Despite a rigorous scale-up of ART access and increased ART coverage in rural KwaZulu-Natal, the proportion of PLWH entering care with AHD remains consistently high, affecting 1 in 4 PLWH presenting to HIV services, and increasingly PLWH who are ART-experienced.

These findings underscore the ongoing importance of CD4 cell count measurement in HIV care, as it plays a crucial role in identifying PLWH who may require additional and effective interventions. Given that male patients were significantly more likely to present with AHD, this highlights the importance of implementing male-friendly services, coupled with intensified adherence support, to effectively address this issue. To further reduce HIV-related mortality, a dual focus on alternative testing strategies to identify PLWH earlier as well as improving retention. Moreover, it remains imperative to identify patients with AHD and provide them with a tailored package of care and services to mitigate the substantial risk of mortality and morbidity, per WHO recommendations.

## Supporting information

**S1 Fig. Predicted mean CD4 counts (cells/µL) of PLWH categorised by year of ART initiation and sex.** Error bars are 95% confidence intervals.
(TIF)

**S2 Fig. Proportion of patients on ART with CD4 < 200 cells/µL by previous ART experience.**
(TIF)

**S3 Fig. Proportion of patients on ART with CD4 < 200 cells/µL by viral load results.**
(TIF)

**S4 Fig. Cumulative Incidence of immune recovery among ART-naïve patients with AHD by sex.**
(TIF)

**S5 Fig. Cumulative Incidence of immune recovery among ART-naïve patients with AHD by Age group.**
(TIF)

## Acknowledgments

We are grateful to the participants and staff of the Eshowe and Mbongolwane ART programme, and the Department of Health of KwaZulu Natal.

## Author contributions

**Conceptualization:** Marcel K. Kitenge, Geoffrey Fatti, Ingrid Eshun-Wilson, Peter S. Nyasulu.

**Data curation:** Marcel K. Kitenge, Ingrid Eshun-Wilson, Peter S. Nyasulu.

**Formal analysis:** Marcel K. Kitenge, Geoffrey Fatti, Ingrid Eshun-Wilson, Peter S. Nyasulu.

**Investigation:** Marcel K. Kitenge, Geoffrey Fatti, Ingrid Eshun-Wilson, Peter S. Nyasulu.

**Methodology:** Marcel K. Kitenge, Geoffrey Fatti, Ingrid Eshun-Wilson, Peter S. Nyasulu.

**Project administration:** Marcel K. Kitenge, Peter S. Nyasulu.

**Resources:** Marcel K. Kitenge, Geoffrey Fatti, Ingrid Eshun-Wilson, Peter S. Nyasulu.

**Software:** Marcel K. Kitenge, Geoffrey Fatti, Ingrid Eshun-Wilson, Peter S. Nyasulu.

**Supervision:** Marcel K. Kitenge, Geoffrey Fatti, Ingrid Eshun-Wilson, Peter S. Nyasulu.

**Validation:** Marcel K. Kitenge, Geoffrey Fatti, Ingrid Eshun-Wilson, Peter S. Nyasulu.

**Visualization:** Marcel K. Kitenge, Geoffrey Fatti, Ingrid Eshun-Wilson, Peter S. Nyasulu.

**Writing – original draft:** Marcel K. Kitenge, Geoffrey Fatti, Ingrid Eshun-Wilson, Peter S. Nyasulu.

**Writing – review & editing:** Marcel K. Kitenge, Geoffrey Fatti, Ingrid Eshun-Wilson, Peter S. Nyasulu.

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
