## [Decision Letter · Decision Letter 0]

25 Mar 2024

PONE-D-24-02035Persistent Advanced HIV Disease in Rural KwaZulu-Natal, South Africa: Trends, Characteristics, and the Urgent Need for Targeted InterventionsPLOS ONE

Dear Dr. Kitenge,

Thank you for submitting your manuscript to PLOS ONE. After careful consideration, we feel that it has merit but does not fully meet PLOS ONE’s publication criteria as it currently stands. Therefore, we invite you to submit a revised version of the manuscript that addresses the points raised during the review process.

We look forward to receiving your revised manuscript.

Kind regards,

Master R.O. Chisale, MRes Medicine (Microbiology)

Academic Editor

PLOS ONE

Journal Requirements:

"NO authors have competing interests"

Reviewers' comments:

Reviewer's Responses to Questions

**Comments to the Author**

1. Is the manuscript technically sound, and do the data support the conclusions?

Reviewer #1: Yes

Reviewer #2: Yes

Reviewer #3: Yes

Reviewer #4: Yes

2. Has the statistical analysis been performed appropriately and rigorously? 

Reviewer #1: Yes

Reviewer #2: Yes

Reviewer #3: I Don't Know

Reviewer #4: Yes

3. Have the authors made all data underlying the findings in their manuscript fully available?

Reviewer #1: No

Reviewer #2: No

Reviewer #3: Yes

Reviewer #4: No

4. Is the manuscript presented in an intelligible fashion and written in standard English?

Reviewer #1: Yes

Reviewer #2: Yes

Reviewer #3: Yes

Reviewer #4: Yes

5. Review Comments to the Author

Reviewer #1: PONE-D-24-02035

Advanced HIV disease and CD4 in KZN 2008 – 2021

Ther authors report the prevalence and patterns of advanced HIV disease (AHD) at presentation to care between 2008 and 2021 in KwaZulu-Natal, South Africa. In addition, the cohort was followed-up to determine factors associated with CD4 recovery after entry into care with AHD. The proportion of people entering care with AHD decreases as the South African antiretroviral therapy (ART) policies become more inclusive. However, the numbers remain high and in later years, participants with AHD are more likely to be ART-experienced, especially men. CD4 recovery can occur in those who return to and remain in care for some time. This is an important topic as HIV services mature; its sampling of a rural population is a strength.

The paper is well-written but with several typos and missing words (lack of attention to detail). The formatting is all over the place. Line numbering only starts in the Discussion and there are no page numbers which has made the review more difficult.

The statistical tests are appropriate and the limitations are described.

General comments:

Please spell out the abbreviations at first use in the abstract, manuscript and tables.

Major comments.

The description of the study design as cross-sectional is a bit misleading as the cohort is followed-up and the analyses applied appropriately for longitudinal data. It would be more accurate to describe a cohort study and present the prevalence of AHD at entry into care (baseline) which is how the cohort is defined.

The description of table 1 is unclear and disorganized. Specify MEDIAN CD4 count. At one point MEAN is presented (S1) but MEDIAN has been used elsewhere. Where are the data on median/mean CD4 presented? In Table S1 why are the means presented when medians are reported in the text? I assume there is a difference between TIME ON ART and FOLLOW-UP TIME in the study. These are confused.

Throughout the analyses when age is used as a categorical variables and 18-24 years is the reference, only data for the 25–31 year category are presented when the effect on the OR/aOR is in the same direction for ALL age categories. it is not clear why 25-31 has been chosen. It could be presented as >24 years.

In all the Tables, please label how the data are presented in the table. E.g., n (%) or median (IQR) etc.

Table 3: I don’t understand the Males/Females with first CD4 test columns.

Line 1: is the CD4 recovery on the ART-naïve or ART-experienced group or both?

Line 6: TB at ART start or TB ever?

The Discussion is overly long and can be condensed. E.g., description of Universal Test and Treat policy; Implications and Future research para 2 repeats para 1

Minor comments

Background, third line: million is duplicated – both the figure and the word are used.

Background further down: ¾ - can’t be right; stage 3 or stage 4

Definitions and outcomes, 3rd sentence: and who experienced a drop…

Statistical analysis, near the end: death and LTFU are competing risks, plural (not a competing risk, singular).

Results, 2nd line: specify/describe the universal test and treat policy. Do not assume the reader is familiar with South African policy. The DATE/S need to be included.

Burden and trends of AHD among ART-experienced patients, line 4: the subject is missing from the sentence. What increased from 4% to 63%?

Before the Table 2 heading there is an incomplete sentence in bold. Why bold? Need to complete sentence.

Line 91: no subject. I think you mean selection bias

References:

Include the url and Access Date for on-line sources

Reviewer #2: The study addresses an important public health issue in a setting with high HIV prevalence. It is well written and the team is commended for that. The folowing are my comments and suggestions

1. In the methods section, why did the authors choose Eshowe and Mbongolwane specifically, beyond HIV rates which are high in the whole of KwaZulu Natal?

2. Justify the use of and assumptions for the mixed effect regression and Poisson models

3. The last sentence here is incomplete: Factors associated with AHD among ART-naïve patients.

In the mixed-effect logistic regression, being male (aOR 1.80; 95% CI 1.71 to 1.90), PLWH

aged 25-31 years (aOR1.65; 95% CI 1.52-1.79) and having TB (AOR 2.46; 95% CI 2.19-2.76)

had higher odds of AHD at ART start (

4. The pages are not numbered

5. Resolve formating issues on multiple pages that are distracting when reading an otherwise well written manuscript

6. The statement: "Finally, having started ART between 2012 and 2015 (aSHR 0.60; 95% CI 0.56 to 0.65) or between 2016 and 2021 (aSHR 0.54; 95% CI 0.49-0.59) compared to having started ART between 2008 and

2011" is incomplete.

7. It would be useful to provide a statistic in place of the words "consistently high" in the opening statement of the discussion.

8. In your discussion of implications for the study, you mention only differentiated care models, but other options are available and may be worth mentioning, such as community support programs, digital adherence tools and so on. There is a need to broaden the options to solve this problem.

Reviewer #3: Background section

Paragraph 1

• Reference 1 does not contain the quoted stats. It might be better to use the WHO country data:

o https://www.who.int/data/gho/data/indicators/indicator-details/GHO/reported-number-of-people-receiving-antiretroviral-therapy

5 588 062 people in SA on ART in 2021

o https://www.who.int/data/gho/data/indicators/indicator-details/GHO/estimated-antiretroviral-therapy-coverage-among-people-living-with-hiv-(-)

75% ART coverage in SA in 2022

• The “Treat all policy” was known as the “Universal Test and Treat policy” in South Africa (2016); it is available at:

http://www.kznhealth.gov.za/test_treat.htm

https://sahivsoc.org/Files/PREP%20and%20TT%20Policy%20-%20Final%20Draft%20-%205%20May%202016%20(HIV%20news).pdf

https://sahivsoc.org/Files/22%208%2016%20Circular%20UTT%20%20%20Decongestion%20CCMT%20Directorate.pdf

These might be better references for implementation of UTT policy in SA.

Background, Paragraph 2

• WHO ¾ stage should be re-written as WHO stage 3 or 4

Setting and participants section

• “…of which only 1.1 million are receiving…”, should be re-phrased as “…1.1 million of whom are receiving…”

• Eshowe and Mbongolwane are mentioned under study setting, but not rationale is given as to why these populations were chosen for the study. Especially if you are drawing conclusion on AHD in rural areas, an attempt should be made to describe the rurality of the setting, and the generalizability to other rural settings.

• No context is given on the relationship between umlalazi municipality (mentioned in the Data Sources section) and Eshowe and Mbongolwane.

Statistical analysis Section

• The use of the Fine-Gray competing risks regression with a proportional sub-distribution hazard (SHR) model should perhaps be references as some readers may be unfamiliar with the method.

Results section, paragraph In the multivariable competing risk regression model of time to CD4 recovery >350 cells/μL following a CD4 <200 cells/μL adjusted for clustering at the health facility, lines 1 to 10.

“These factors were associated with a reduced risk of CD4 recovery to >350 cells/μL.” It is unclear what factors you are referring to, as all above-mentioned factors has a better CD4 > 350 recovery.

There was also a better CD4 > 350 recovery on non-stavudine containing regimens, which is not mentioned here.

Table 1:

• “Stavudine” is misspelled

• “ART start” should be changed to “ART initiation” for consistency of terminology.

Discussion Section, line 12 to 19

• The description of AHD in the rural setting, should be contrasted with the existing literature about urban settings.

Discussion Section, line 30

• “September 2015” should be corrected to “2016”.

The SA DoH guideline of 2015 set the CD4 threshold at 500, available here:

https://sahivsoc.org/Files/ART%20Guidelines%2015052015.pdf

• The other factors besides gender, which affect CD4 recovery i.e. age group, ART regimen, time period in which ART was initiated, being on TB treatment – also warrant further discussion.

Figure 2

Suggestion: Add overlying labels on the figure, showing the changes in national policy on CD4 threshold at ART initiation. The decline in AHD does follow the policy changes from CD4 of 200 in 2008, to 350 in 2013, to 500 in 2014, then to UTT in 2016.

Sahivsoc.org has the old DOH policies, if you are looking for them.

S1 Appendix.

• The title: Predicted average relative risk of Males with First CD4 count <200 cells/mL

by age group. Does not match the info on the graph.

The y-axis on the graph should specify whether that is the CD4 at initiation or whether it is the change in CD4.

S2 appendix and S3 appendix

• It is unclear what is being demonstrated here. Perhaps these percentages should be accompanied by absolute numbers. Or another analysis should be done - % of people on ART with no interruption, and VL <1000 with a CD4<200, over time.

This will tell you whether there is a truly increasing burden of failure of immune recovery while on effective ART, (which I highly doubt there was)

The data is the current format don’t give that information, as there was an increasing number of people on ART over time. So the proportion in that category will logically increase over time.

Reviewer #4: Thank you for the opportunity to review this manuscript on advanced HIV disease in a rural population in KZN. The manuscript describes an important topic around which renewed focus is needed, and these are valuable data to add to the evidence body.

The presentation and readability of the results section of the manuscript could be improved.

A few more specific comments below

Background

Amend “WHO ¾ stage” to read “WHO stage 3 / 4” or “3 or 4”

Methods

Your inclusion criteria state “Our study included all individuals

aged 18 years and older living with HIV who were receiving ART, and who had undergone

baseline and subsequent CD4 cell measurements between January 2008 and June 2021.”

Please define what baseline is in this case.

Although it is mentioned in the limitations of the study, it would be useful to include in the results (in Figure 1) the numbers of adults on ART who were excluded because they did not have a CD4 cell count at ART initiation and the numbers without a subsequent CD4 measure. With the introduction of treat all CD4 testing has declined, and this could introduce bias into the results, since those who receive CD4 testing are sicker. It would help the reader assess the potential for bias in your results.

Last sentence under ‘Data sources and variable’ is missing a word.

Results

The title for Figure S1 is incorrect

Table 3 may be better presented as a graph

Second last sentence of results is incomplete: “Finally, having started ART between 2012 and 2015 (aSHR 0.60; 95% CI 0.56 to 0.65) or between 2016 and 2021 (aSHR 0.54; 95% CI 0.49-0.59) compared to having started ART between 2008 and 2011.”

It may be better to summarize the factors associated with reduced risk of CD4 recovery without providing the aSHRs in the text, to improve readability.

Please clarify what prior attendance unknown means?

Results show later year of ART start reduced risk of AHD on ART, could this be as a result of less follow-up time? Would be worth adding to the discussion about this result

Discussion

More discussion about the ART-experienced patients with AHD would be beneficial. From your Figure S2 it looks like an increasing proportion are in care (ART experienced, regular attendance) and not treatment interrupters, and the majority (69%) were virally suppressed. Some further clarity on these results is needed in the discussion.

6. PLOS authors have the option to publish the peer review history of their article (what does this mean? ). If published, this will include your full peer review and any attached files.

**Do you want your identity to be public for this peer review?** For information about this choice, including consent withdrawal, please see our Privacy Policy .

Reviewer #1: No

Reviewer #2: **Yes: ** Gershom Chongwe

Reviewer #3: **Yes: ** Kerusha Govender

Reviewer #4: No

---

## [Author Response · Author response to Decision Letter 1]

31 Jul 2024

Dear reviewers,

Re: Responses to reviewers’ comments

Please find below a point-by-point response to the issues raised by the reviewers. The changes made to the manuscript are highlighted in track changes in the revised version.

Original Title: Persistent Advanced HIV Disease in Rural KwaZulu-Natal, South Africa: Trend, Characteristics and Urgent Need for Targeted Interventions.

Reviewer 1

General Comment

The authors report the prevalence and patterns of advanced HIV disease (AHD) at presentation to care between 2008 and 2021 in KwaZulu-Natal, South Africa. In addition, the cohort was followed up to determine factors associated with CD4 recovery after entry into care with AHD. The proportion of people entering care with AHD decreases as the South African antiretroviral therapy (ART) policies become more inclusive. However, the numbers remain high and in later years, participants with AHD are more likely to be ART-experienced, especially men. CD4 recovery can occur in those who return to and remain in care for some time. This is an important topic as HIV services mature; its sampling of a rural population is a strength.

The paper is well-written but with several typos and missing words (lack of attention to detail). The formatting is all over the place. Line numbering only starts in the Discussion and there are no page numbers which has made the review more difficult.

General comments:

Please spell out the abbreviations at first use in the abstract, manuscript and tables.

Response

We express our gratitude to the reviewer for providing valuable feedback, and we acknowledge the validity of his comment. We sincerely apologize for the typos , missing words, and absence of line numbering. Consequently, we have diligently rectified this oversight.

Additionally, we have spelled out the abbreviations throughout the manuscript. For example see line 31 and 168.

Major comments

The description of the study design as cross-sectional is a bit misleading as the cohort is followed up and the analyses applied appropriately for longitudinal data. It would be more accurate to describe a cohort study and present the prevalence of AHD at entry into care (baseline) which is how the cohort is defined.

Response

We thank the reviewer for the comment. We omitted “serial” in line 30. Although we used longitudinal data, the study is a serial cross-sectional study as we aim to determine the prevalence of advanced HIV disease from 2008 to 2021, describe trends in advanced HIV disease in patients living with HIV across different time points, and investigate risk factors associated with advanced HIV disease in both ART-naïve and ART-experienced patients.

This has been adequately changed and now reads as follows in lines 30-31: “A serial cross-sectional analysis of annual CD4 count distribution was conducted among ART patients aged 18 years and older”

Comment

The description of Table 1 is unclear and disorganized. Specify MEDIAN CD4 count. At one point MEAN is presented (S1) but MEDIAN has been used elsewhere. Where are the data on median/mean CD4 presented? In Table S1 why are the means presented when medians are reported in the text? I assume there is a difference between TIME ON ART and FOLLOW-UP TIME in the study. These are confused.

Response

We thank the reviewer for the comment. As indicated in the statistical analysis “Continuous variables were summarised using mean and standard deviation if normally distributed, otherwise median, and interquartile range (IQR) were used…” a shown in line 153-154. For example, age was not normally distributed, so we used the median and interquartile range. However, for CD4, which was normally distributed, we used the mean. As such we have aligned the text with S1 Appendix and can be seen in lines 155-156, where it reads as follows: “The mean CD4 count increased across years, the largest increase in CD4 counts was observed in 2016…”

Additionally, as you suggested and to avoid confusion, we have changed “Follow-up on ART” to “Time on ART”. See Table 1.

Comment

Throughout the analyses when age is used as a categorical variable and 18-24 years is the reference, only data for the 25–31 year category are presented when the effect on the OR/aOR is in the same direction for ALL age categories. it is not clear why 25-31 has been chosen. It could be presented as >24 years.

Response

We thank the reviewer for the comment. We have made amendment as suggested. This is shown in line 195-199 and this reads as follows “In the mixed-effect logistic regression, being male (Adjusted odds ratio[aOR] 1.80; 95% CI 1.71 to 1.90), PLWH in the age groups of 25-31 years, 32-38 years, and 39-45 years exhibited a higher odds of AHD at ART initiation when compared to those aged 18-24 years. The aOR for these age groups were 1.65 (95% CI 1.52 to 1.79), 2.15 (95% CI 1.98 to 2.34), and 2.22 (95% CI 2.03 to 2.44), respectively.”

Additionally, we followed the guidelines provided by the International Journal of Epidemiology on forming categories, which state, “Age grouping should be mid-decade to mid-decade or five-year age groups.” This approach has been applied in this manuscript for the categorisation of age.

Comment

In all the Tables, please label how the data are presented in the table. E.g., n (%) or median (IQR) etc.

Response

We appreciate the reviewer's suggestion. We have added the missing labels to both Table 1 and Table 2.

Comment

Table 3: I don’t understand the Males/Females with first CD4 test columns.

Line 1: is the CD4 recovery on the ART-naïve or ART-experienced group or both?

Line 6: TB at ART start or TB ever?

Response

We appreciate the reviewer’s suggestion. We have adjusted Table 3 by separating the data for males, females, and the total. In table 4 we have expanded the title and this reads as follows “Table 1. Factors associated with CD4 recovery to >350 cells/μL following a CD4 <200 cells/μL from a competing risks regression model among ART-naïve and ART-experienced patients”. We have additionally expanded on this in lines 238-240 and it now reads as follows “Time to CD4 recovery to >350 cells/μL following a CD4 <200 cells/μL and outcomes post-identification of CD4 <200 cells/μL among ART-naïve and ART-experienced patients”.

Comment

The Discussion is overly long and can be condensed. E.g., description of Universal Test and Treat policy; Implications and Future research para 2 repeats para 1

Response

To shorten the discussion section, we have condensed and removed some repetitive paragraphs. See lines 298-302.

Comment

Minor comments

Background, third line: million is duplicated – both the figure and the word are used.

Response

We thank the reviewer for the comment. This has been adequately removed and It reads as follows “By 2021, it had reached 74% of all individuals infected with the human immunodeficiency virus (HIV), with 5,588,062 adults on ART”. See line 56-58.

Comment

Background further down: ¾ - can’t be right; stage 3 or stage 4

Response

We thank the reviewer for the comment. We have adequately addressed this and it reads as follows “Conversely, a CD4 count below 200 cells/μL or WHO stage 3 or 4 event (Advanced HIV Disease) strongly predicts severe morbidity, comorbidities such as cryptococcal meningitis and Tuberculosis (TB), as well as mortality”. See line 65-66.

Comment

Definitions and outcomes, 3rd sentence: and who experienced a drop…

Statistical analysis, near the end: death and LTFU are competing risks, plural (not a competing risk, singular).

Response

We thanks the reviewer for the comment. As shown in lines 132-133, we have added the missing word, and it now reads as follows: “ART-experienced patients were those who entered care with CD4 count ≥ 200 cells/μL and who experienced a drop in CD4 count to <200 cells/μL during follow-up...”. Additionally, we have addressed misspelling of competing risks throughout the manuscript.

Comment

Results, 2nd line: specify/describe the universal test and treat policy. Do not assume the reader is familiar with South African policy. The DATE/S need to be included.

Response

We appreciate the reviewer’s suggestion. We have removed the UTT. However, details on the Universal test and treat policy implementation in South Africa ins presented under ‘study settings and participants’. See line 97-101 and this reads as follows “South Africa implemented the Universal Test and Treat (UTT) on September 1, 2016. Under this strategy, all public health facilities are required to scale up ART initiation for all HIV-infected individuals according to UTT guidelines, with an emphasis on providing same-day initiation for individuals newly diagnosed with HIV who are clinically and psychologically ready for lifelong ART(16)”

Comment

Burden and trends of AHD among ART-experienced patients, line 4: the subject is missing from the sentence. What increased from 4% to 63%.

Response

We thank the reviewer for the comment. We have addressed this and it reads as follows “The proportion of patients with AHD who were ART-experienced increased from 4% in 2008 to 63% in 2021”. See line 41-42.

Comment

Before the Table 2 heading there is an incomplete sentence in bold. Why bold? Need to complete sentence.

Response

We thank the reviewer for the comment. we have completed Table 2 heading and it reads as follows “Table 2. Factors associated with AHD among ART-naive patients at initiation of ART”. This also is reflected in the text as shown in line 200-201. Additionally, we have removed the bold typeface for all headings for tables and figures as suggested.

Comment

Line 91: no subject. I think you mean selection bias

Response

We thank the reviewer for the comment. We have added the missing word as shown in line 375-381 and this reads as follows “This may have introduced selection bias and may limit the applicability of these findings to the entire population enrolled in the HIV program.”

Reviewer 2

The study addresses an important public health issue in a setting with high HIV prevalence. It is well written and the team is commended for that. The following are my comments and suggestions

Comment

In the methods section, why did the authors choose Eshowe and Mbongolwane specifically, beyond HIV rates which are high in the whole of KwaZulu Natal?

Response

We thank the reviewer for the comment. Eshowe is a small town that lies in the uMlalazi Municipality, King Cetshwayo district in KwaZulu Natal, South Africa.

We selected these areas primarily because of the high incidence and prevalence of HIV and TB compared to the rest of South Africa. The overall adult HIV prevalence in the area is 26.4% among adults aged 15-59 years. Additionally, between and 2019, Médecins Sans Frontières and the Department of Health implemented numerous community-based HIV/TB interventions. Following the implementation of the se interventions, a population-based survey conducted by MSF, these areas exceeded the UNAIDS 90-90-90 targets by achieving 90-94-95 in 2018. Despite this major achievement, programmatic data showed that the prevalence of AHD was still higher with 1 in 4 PLWH presenting to HIV care. Conclusions drawn from this study will be generalizable to other rural settings, especially those with high HIV prevalence and low-income.

Some of this information has been added under the study setting, as shown in lines 101-105 and 102-120, and it reads as follows: “Since 2011, Médecins Sans Frontières (MSF) and the Department of Health (DoH) of KwaZulu-Natal have been implementing a community-based HIV/TB project. The project involves 10 primary health facilities and two hospitals in the Eshowe and Mbongolwane areas. The areas are predominantly rural areas with one semi-urban market town, with a combined population of 114,490 in 2021 (18). According to a population-based survey conducted by MSF, these areas exceeded the UNAIDS 90-90-90 targets by achieving 90-94-95 in 2018. The survey also revealed that HIV prevalence in the study areas was an estimated 26.4 among 15-59 years old adults (15)”.

It is against this background, we conducted this study solely aiming at determining the burden of and treatment outcomes of advanced HIV disease among patients starting ART and those ART-experienced in the Eshowe and Mbongolwane areas.

Comment

2. Justify the use of and assumptions for the mixed effect regression and Poisson models

Response

We thank the reviewer for the comment. The primary reason we used mixed effect logistic regression is that we assumed a random effect for each facility. Given that the study was conducted at 10 public health facilities, we used mixed effect logistic regression to model binary outcomes variables (assessing the probability of having AHD among ART-naïve patients at ART initiation) , in which the odds of the outcomes are modelled as linear combination of the predictor variables when data from these facilities are clustered. Additionally, given some CD4 data were missing. Mixed-effect logistic regression models are mostly based on the assumption that missing data are random and non-informative.

While, Poisson regression model allows to assess epidemiological studies when the main outcome is an integer. It assumes that these types of outcomes are random variables with positive integers and are distributed as the Poisson probability distribution . The Poisson regression model is used in longitudinal cohort to estimate the expected value of outcomes among exposure and non-exposure groups, adjusted for the effect of potential confounders. One advantage of this model is that it can be used to obtain an estimate of the relative risk (RR), adjusted for potential confounders. Based on the aforementioned reasons, we opted to utilize both mixed-effect logistic regression models and Poisson regression models. The former was employed to determine the odds of AHD among ART-naïve patients at ART initiation, adjusted for age, sex, calendar year, and TB status. Meanwhile, the latter was utilized to identify predictors of AHD among ART-experienced patients, with relative risk serving as the measure of effect.

Comment

The last sentence here is incomplete: Factors associated with AHD among ART-naïve patients.

In the mixed-effect logistic regression, being male (aOR 1.80; 95% CI 1.71 to 1.90), PLWH

aged 25-31 years (aOR1.65; 95% CI 1.52-1.79) and having TB (AOR 2.46; 95% CI 2.19-2.76)

had higher odds of AHD at ART start.

Response

We appreciate the reviewer’s comment. We have completed the sentence as shown in line 195-201 and it now rears as follows “In the mixed-effect logistic regression, being male (Adjusted odds ratio[aOR] 1.80; 95% CI 1.71 to 1.90), PLWH in the age groups of 25-31 years, 32-38 years, and 39-45 years exhibited a higher odds of AHD at ART initiation when compared to those aged 18-24 years . The aOR for these age groups were 1.65 (95% CI 1.52 to 1.79), 2.15 (95% CI 1.98 to 2.34), and 2.22 (95% CI 2.03 to 2.44), respectively. PLWH aged 25-31 years (aOR1.65; 95% CI 1.52-1.79) and Having TB (AOR 2.46; 95% CI 2.19-2.76) was associated with higher odds of AHD at ART initiation (Table 2).”

Comment

The pages are not numbered

Response

We thank the reviewer for the comment and apologize for this omission. We have added the page number and line numbers.

Comment

Resolve formatting issues on multiple pages that are distracting when reading an otherwise well written manuscript

Response

We appreciate the reviewer for the comment and apologize for the formatting issues. We have resolved all formatting issues at the best of our ability.

Comment

The statement: "Finally, having started ART between 2012 and 2015 (aSHR 0.60; 95% CI 0.56 to 0.65) or between 2016 and 2021 (aSHR 0.54; 95% CI 0.49-0.59) compared to having started ART between 2008 and 2011" is incomplete.

Response

We appreciate the reviewer’s comment. We have completed the sentence as shown in line 258-261 and it now rears as follows “Starting ART between 2012 and 2015 (aSHR 0.60; 95% CI 0.56 to 0.65) or between 2016 and 2021 (aSHR 0.54; 95% CI 0.49 to 0.59) compared to starting ART between 2008 and 2011 was associated with a reduced relative incidence of CD4 recovery to >350 cells/μL.”

Comment

It would be useful to provide a statistic in place of the wor

---

## [Decision Letter · Decision Letter 1]

4 Oct 2024

PONE-D-24-02035R1

Persistent Advanced HIV Disease in Rural KwaZulu-Natal, South Africa: Trends, Characteristics, and the Urgent Need for Targeted Interventions

PLOS ONE

Dear Dr.  Kitenge,

Thank you for submitting your manuscript to PLOS ONE. After careful consideration, we feel that it has merit but does not fully meet PLOS ONE’s publication criteria as it currently stands. Therefore, we invite you to submit a revised version of the manuscript that addresses the points raised during the review process.

I want to congratulate you all for trying to respond to the editor/peer reviewers’ comments and suggestions. This is a n excellent finding of that could be an input for province/national HIV policy. Yet, there are some remaining constructive suggestions: - pay attention to reducing the size of the article through summarizing and trimming sections; and sticking to PLOSE ONE journal structure/requirement.

We look forward to receiving your revised manuscript.

Kind regards,

Zewdu Gashu Dememew, M.D

Academic Editor

PLOS ONE

Journal Requirements:

Additional Editor Comments:

Dear Authors,

I want to congratulate you all for trying to respond to the editor/peer reviewers’ comments and suggestions. This is a n excellent finding of that could be an input for province/national HIV policy.

Yet, there are some remaining constructive suggestions: - pay attention to reducing the size of the article through summarizing and trimming sections; and sticking to PLOSE ONE journal structure/requirement.

Best,

Reviewers' comments:

Reviewer's Responses to Questions

**Comments to the Author**

1. If the authors have adequately addressed your comments raised in a previous round of review and you feel that this manuscript is now acceptable for publication, you may indicate that here to bypass the “Comments to the Author” section, enter your conflict of interest statement in the “Confidential to Editor” section, and submit your "Accept" recommendation.

Reviewer #1: All comments have been addressed

Reviewer #3: All comments have been addressed

Reviewer #4: (No Response)

2. Is the manuscript technically sound, and do the data support the conclusions?

Reviewer #1: Yes

Reviewer #3: Yes

Reviewer #4: Partly

3. Has the statistical analysis been performed appropriately and rigorously? 

Reviewer #1: Yes

Reviewer #3: Yes

Reviewer #4: Yes

4. Have the authors made all data underlying the findings in their manuscript fully available?

Reviewer #1: No

Reviewer #3: Yes

Reviewer #4: No

5. Is the manuscript presented in an intelligible fashion and written in standard English?

Reviewer #1: Yes

Reviewer #3: Yes

Reviewer #4: No

6. Review Comments to the Author

Reviewer #1: Thank you for the revised manuscript. My comments have been adequately addressed.

The formatting is still problematic but I assume this will be sorted out during production.

Line 62: men living with HIV

Line 63: typo HIB = HIV

Supporting Information 1st graph: heading includes CD41

Reviewer #3: All reviewer comments have been sufficiently addressed.

I have no additional comments

Reviewer #4: Many thanks for the opportunity to re-review this manuscript. I recommend that this manuscript undergo further editing to improve readability.

Below are my further comments:

• I agree with the other reviewers regarding their suggestions for the age groupings used in this analysis. The authors response does not appear to justify their choice of age-groupings either. They reference their choice as being based on the international journal of epidemiology guidelines which are: “age grouping should be mid-decade to mid-decade or in five-year age groups (e.g. 35-44 or 35-39, 40—44, etc, but not 20-29, 30-39 or other groupings).” The authors have chosen 7-year age bands (25-31, 32-38, 39-45) which are not in line with this recommendation. I recommend amending the analysis to be in line with these guidelines.

• Based on my recommendation the authors added the definition of prior attendance unknown to the methods section: “We also defined prior attendance unknown if the patients had neither a ≥90-day interruption in care nor regular attendance or continuously in care.” I feel this sentence can be clarified further. It implies that “regular attendance” is different from “continuously in care”, and neither of these terms have been defined. I feel the authors should explicitly state that they had a variable for patient attendance and state and define each of the 3 categories.

• Results section where the factors associated with AHD among ART-naïve patients are described could be further edited for clarity and readability

• Suggest rephrasing the term “during ART” to “after ART initiation”, since it includes PLWH who interrupt treatment.

• If I understand correctly the AHD among ART-experienced group includes only patients who initiated ART with a CD4>200. The results which describe these patients seem to suggest that the majority (58%) were continuously in care, and virally suppressed (69%).

o In the discussion it is stated: “there was an increase in the number of PLWH with AHD who were ART-experienced (PLWH who have previously initiated ART and are re-engaging with AHD after a period without effective ART or failed therapy).” This definition in parenthesis does not agree with what was described in the results (that the majority were continuously in care, and virally suppressed) nor does it match the definition that was stated in the methods section: “ART-experienced patients were those who entered care with CD4 count ≥ 200 cells/μL and who experienced a drop in CD4 count to <200 cells/μL during follow-up (exposure to ART for at least 6 months or more), including those lost to follow-up and who returned to care.” I suggest removing the definition in brackets from the discussion.

o I also suggest removing the sentence: “Of these AHD patients who were ART-experienced, 27% had either had a prior attendance unknown or interrupted treatment for 90 days or more and 14% had a detectable VL of ≥ 1000 copies/ml, suggesting either adherence problems or failed therapy.”

o I suggest adding to the subsequent sentence: “Additionally, among those ART-experienced patients with AHD with documented VL, 69% were virally suppressed.”, that 58% were continuously in care.

o The authors correctly include a discussion about AHD and cycling in and out of care as there is much literature to support this. But your results do not agree with this statement. The authors also say “These findings underscore the necessity for strategies aimed at closing gaps in HIV care, promoting re-engagement interventions (48), and encouraging individuals to return before experiencing significant deterioration (49, 50).” I agree with these sentiments, but the study’s results do not show this since the majority of those with AHD after ART initiation were continuously in care and virally suppressed. While the authors have now included a sentence about patients who experience AHD despite ART and VL suppression, the rest of the paragraph needs to be reframed to be in line with the results. The authors reference the ZIMPHIA survey, but in that study only 30% of patients with advanced disease were virally suppressed. The authors have not discussed the much higher proportion of 69% that was found in this study or possible reasons for this.

o Reference 44 and 45 which are meant to support the statement do not seem to be appropriate: “The subset of PLWH experiencing AHD despite prolonged ART and viral load suppression has been noted in several other studies from the region (43-45).” The Ousley study is of hospitalized patients and found that between 81% and 84% of patients admitted to hospital who were ART experienced also had a detectable VL of >1000 copies/mL. This study’s findings seem to be exactly opposite to the point the authors are trying to make. The Nanzigu study (ref 45) describes immune recovery and offers no evidence about advanced disease and viral suppression.

7. PLOS authors have the option to publish the peer review history of their article (what does this mean? ). If published, this will include your full peer review and any attached files.

**Do you want your identity to be public for this peer review?** For information about this choice, including consent withdrawal, please see our Privacy Policy .

Reviewer #1: No

Reviewer #3: **Yes: ** Kerusha Govender

Reviewer #4: No

---

## [Author Response · Author response to Decision Letter 2]

27 Oct 2024

27 October 2024

Dear reviewers,

Re: Responses to reviewers’ comments

Please find below a point-by-point response to the issues raised by the reviewers. The changes made to the manuscript are highlighted in track changes in the revised version.

Original Title: Persistent Advanced HIV Disease in Rural KwaZulu-Natal, South Africa: Trend, Characteristics and Urgent Need for Targeted Interventions.

General Comment

General comments and suggestions

Congratulation for the authors for trying to respond to the editor/peer reviewers’ comments and suggestions. It is well done. Yet, there are some remaining constructive suggestions or comments to deal with to further enrich the article. Please pay attention to reducing the size of the article through summarizing, merging and trimming sections; English language edits; and sticking to PLOSE ONE journal structure/requirement. Submission Guidelines | PLOS ONE.

Also, the authors are anticipated to clearly describe and align the study design and analysis.

Otherwise, this is a n excellent finding of ADH in high HIV burden setting of South Africa that could be an input for province/national HIV policy.

Response

We express our gratitude to the reviewer for their valuable feedback and acknowledge the validity of the comment. We have reduced the length of the manuscript by summarizing, merging, and trimming down certain sections throughout the manuscript.

Comment 1

Specific comments and suggestions

KEY Words: check if Advanced HIV disease, AHD are to be retained-- both seem similar. Check if KwaZulu-Natal could be part of the key words.

Response

We appreciate the reviewer's suggestion. We have deleted AHD and added KwaZulu-Natal part of the keywords, and this is shown in line 20-21 and it reads as follows “keywords: Disease burden, Advanced HIV disease, ART-naïve, ART-experienced, KwaZulu-Natal, South Africa.”

Comment 2

Abstract

Check if PLWH is the right/standard abbreviation for persons living with HIV as compared to PLHIV

Response

Thank you to the reviewer for the comment. We have replaced "PLWH" with "PLHIV" throughout the manuscript.

Comment 3

A serial cross-sectional analysis’ is aligned with” FineGreys competing risks regression with proportional sub-distribution hazard models…’’ analysis. PLEASE review this issue again! Literally, this is a retrospective cohort study using the routine data (of ~13 years), as rightly mentioned under Ethical statement as,”…the study was retrospective in nature…”. Therefore, the burden of ADH could be described as both prevalence and incidence. The terms, ‘’trend, risk regression, hazard models, cumulative incidence, 100 adult-years follow-up time … ‘’ are usual used with cohort/longitudinal study designs.

Response

Thank you to the reviewer for the suggestion. We have revised this to "retrospective cohort," as reflected in lines 31 and 93, which now read as follows: “This was a retrospective cohort design of annual CD4 cell count data among adults receiving ART between January 1, 2008, and June 30, 2021, using routinely collected ART program data at 10 public health facilities.”

Comment 4

Check if you need to add ’ medial’ before ‘’ … CD4 count was 277…’’.

Response

We thank the reviewer for the suggestion. We have added median as shown in line 171-173 and it reads as follows “A total of 34,729 patients were included of which 68.5% were females. The median age of the study sample was 33.5 years (interquartile range [IQR] 27-41 years), and the median CD4 count was 277 cells/μL (IQR, 149-452 cells/μL).”

Comment 5

Check if the statement, “Across all periods, those entering care with AHD were more likely to be men when compared to women” needs to be substantiated with statistically significant levels in the bracket.

Check punctuation and spacing in,” In addition,” and:” ART-experienced”.

Response

Thank you for the reviewer’s insightful comment. We have added supporting statistical information to strengthen the statement, as detailed in line 42-43 and it reads as follows “Across all periods, those entering care with AHD were more likely to be men when compared to women (Relative risk [RR] 1.49 ; 95% 1.33 to 1.67). In addition, the proportion of patients with AHD who were ART-experienced increased from 4% in 2008 to 63% in 2021.”

Comment 6

The authors should have line numbered all the document. No page numbers as well.

Response

We appreciate the reviewer’s comment, we have expanded the line number to all pages and have added number

Comment 7

Introduction section

“Disease for those ART-naïve or returning to care after interruption in treatment) strongly predicts severe morbidity, comorbidities such as cryptococcal meningitis and Tuberculosis (TB), as well as mortality (11, 12). “

Check if this could be shortened to “Disease for those ART-naïve or returning to care after interruption in treatment strongly predicts severe morbidity, comorbidities and mortality (11, 12). “

Response

We thank the reviewer for the suggestion. The sentence has been revised as per the recommendation on line 70, now reading as follows “…Disease for those ART-naïve or returning to care after interruption in treatment) strongly predicts severe morbidity, comorbidities and mortality”

Comment 8

Methods section

Check and correct the spacing in,” HIV-infected”, “population-based”, “ART-experienced”. Please do same throughout the document.

Response

We appreciate the reviewer’s comment. These have been revised as per the recommendation.

Comment 9

The sentence,” KwaZulu-Natal, South Africa has 1.9 million PLWH, 1.1 million of whom are receiving ART (17).” is not clear. Check if it could be written as, “KwaZulu-Natal province in South Africa has 1.9 million PLWH, 1.1 millions of whom are receiving ART in xxxx (year/period) (17).’

Response

We thank the reviewer for the suggestion. The sentence has been revised as per the recommendation on line 101-102, now reading as follows “KwaZulu-Natal province in South Africa has 1.9 million PLWH, 1.1 millions of whom were receiving ART in 2023.”

Comment 10

Result section –well narrated yet needs to be condensed/summarized.

Please review this sentence, ”… and Having TB (AOR 2.46; 95% CI 2.19-2.76) was associated with higher odds of AHD at ART initiation (Table 2).”

Response

We thank the reviewer for the comment. We have condensed the results section as reflected in lines 170-177, 179-238, and 279-294.

Comment 11

Check if heading write up fits to PLOS ONE criteria. E.g 3.1 Discussion, 3. Results ….

Response

We appreciate the insightful input and apologize for the oversight. This has been addressed appropriately, as shown in line 332.

Comment 12

Discussion section:-The interpretation of the finding and the suggestion is well narrated.

Please restructure the discussion paragraph by paragraph (P).

Example: check if this structuring make sense--e.g P1:17-24, P2: 25-43 P3:44-56 P4:57-67 PY5:67-73 PY6: 73-79 PY7: 80-88

Also try to compare and contrast the findings with previous similar studies: For instance, ART-experience and ADH,

Response

We thank the reviewer for the comment. We have restructured the discussion section paragraph by paragraph and trimmed it as suggested. These changes are reflected in lines 348-369, 378-441, and 448-463.

Comment 13

Make similar decimal point description of %. E.g line # 25-30: 36.6%, 62%, 43.45%. These used different decimal points. Suggested to write with same decimal point throughout.

Response

We thank the reviewer for the comment. We have addressed these inconsistencies in decimal point reporting as shown in line 373 and 377.

Comment 14

limitation:

Refer to PLOS ONE journal requirements while narrating this section-which should be condensed/summarized/trimmed

Response

We appreciate the reviewer’s comment. We have condensed and summarized this section as recommended, as shown in lines 489-504.

Comment 15

Line # 100: you may replace ‘conclusion’ with “interpretation”.

Response

We thank the reviewer for this comment, and we have replaced “conclusion” with “interpretation”. This change is shown in line 492-493 as it reads as follows “However, interpretation should be done with caution due to limitations inherent in observational study designs.”

Comment 16

Implications for practice and future research

The content is very important. Yet need to be summarized and be part of discussion above. Again, please refer to PLOS ONE journal structure : Submission Guidelines | PLOS ONE

Response

We appreciate the reviewer’s comment. We have removed implication for practices and future research from the manuscript.

---

## [Decision Letter · Decision Letter 2]

3 Jan 2025

Persistent Advanced HIV Disease in Rural KwaZulu-Natal, South Africa: Trends, Characteristics, and the Urgent Need for Targeted Interventions

PONE-D-24-02035R2

Dear Dr. Kitenge,

We’re pleased to inform you that your manuscript has been judged scientifically suitable for publication and will be formally accepted for publication once it meets all outstanding technical requirements.

Kind regards,

Matthew J. Mimiaga, ScD, MPH

Academic Editor

PLOS ONE

Additional Editor Comments (optional):

I have facilitated the peer-review of your manuscript, “Persistent Advanced HIV Disease in Rural KwaZulu-Natal, South Africa: Trends, Characteristics, and the Urgent Need for Targeted Interventions.” Overall, the reviewers were enthusiastic about this well-conducted study on trends in annual CD4 count distribution and to characterize adult persons living with HIV (PLWH) on ART who have AHD in rural KwaZulu-Natal, South Africa. All three reviewers found your paper to be of great public health significance and were enthusiastic about its publication in PLOS One. Based on their expert reviews, I am in agreement.

Reviewers' comments:

Reviewer's Responses to Questions

**Comments to the Author**

1. If the authors have adequately addressed your comments raised in a previous round of review and you feel that this manuscript is now acceptable for publication, you may indicate that here to bypass the “Comments to the Author” section, enter your conflict of interest statement in the “Confidential to Editor” section, and submit your "Accept" recommendation.

Reviewer #1: All comments have been addressed

Reviewer #3: All comments have been addressed

Reviewer #4: (No Response)

2. Is the manuscript technically sound, and do the data support the conclusions?

Reviewer #1: Yes

Reviewer #3: Yes

Reviewer #4: Yes

3. Has the statistical analysis been performed appropriately and rigorously? 

Reviewer #1: Yes

Reviewer #3: Yes

Reviewer #4: Yes

4. Have the authors made all data underlying the findings in their manuscript fully available?

Reviewer #1: (No Response)

Reviewer #3: Yes

Reviewer #4: No

5. Is the manuscript presented in an intelligible fashion and written in standard English?

Reviewer #1: No

Reviewer #3: Yes

Reviewer #4: Yes

6. Review Comments to the Author

Reviewer #1: (No Response)

Reviewer #3: All comments have been addresssed.

The manuscript is well-presented, and the research has significant implications.

Reviewer #4: Thanks for the opportunity to give input into this manuscript again. I think my comments following the first revision to this manuscript were not seen by the authors as there is no response to these recorded so I have included those that still need to be addressed below:

• I agree with the other reviewers regarding their suggestions for the age groupings used in this analysis. The authors response does not appear to justify their choice of age-groupings either. They reference their choice as being based on the international journal of epidemiology guidelines which are: “age grouping should be mid-decade to mid-decade or in five-year age groups (e.g. 35-44 or 35-39, 40—44, etc, but not 20-29, 30-39 or other groupings).” The authors have chosen 7-year age bands (25-31, 32-38, 39-45) which are not in line with this recommendation. I recommend amending the analysis to be in line with these guidelines.

• Based on my recommendation the authors added the definition of prior attendance unknown to the methods section: “We also defined prior attendance unknown if the patients had neither a ≥90-day interruption in care nor regular attendance or continuously in care.”I feel this sentence can be clarified further. It implies that “regular attendance” is different from “continuously in care”, and neither of these terms have been defined. I feel the authors should explicitly state that they had a variable for patient attendance and state and define each of the 3 categories.

• Results section – Line 163-164 – clearly state or clarify that CD4 at ART initiation rose during the period and not all CD4 counts

• Results section where the factors associated with AHD among ART-naïve patients are described could be further edited for clarity and readability

o Line 168-170 – could be reduced to “being male, those with older age, and those with tuberculosis”, with aORs listed in brackets for each. It is not clear why the aOR for the >45 age group is omitted and all other age groups mentioned

• Suggest rephrasing the term “during ART” to “after ART initiation”, since it includes PLWH who interrupt treatment.

• If I understand correctly the AHD among ART-experienced group includes only patients who initiated ART with a CD4>200. The results which describe these patients seem to suggest that the majority (58%) were continuously in care, and virally suppressed (69%).

o In the discussion (line 57-59) it is stated: “there was an increase in the number of PLWH with AHD who were ART-experienced (PLWH who have previously initiated ART and are re-engaging with AHD after a period without effective ART or failed therapy).” This definition in parenthesis does not agree with what was described in the results (that the majority were continuously in care, and virally suppressed) nor does it match the definition that was stated in the methods section: “ART-experienced patients were those who entered care with CD4 count ≥ 200 cells/μL and who experienced a drop in CD4 count to <200 cells/μL during follow-up (exposure to ART for at least 6 months or more), including those lost to follow-up and who returned to care.” I suggest removing the definition in brackets from the discussion.

o I also suggest removing the sentence (line 60-62): “Of these AHD patients who were ART-experienced, 27% had either had a prior attendance unknown or interrupted treatment for 90 days or more and 14% had a detectable VL of ≥ 1000 copies/ml, suggesting either adherence problems or failed therapy.”

o I suggest adding to the subsequent sentence: “Additionally, among those ART-experienced patients with AHD with documented VL, 69% were virally suppressed.”, that 58% were continuously in care.

o The authors correctly include a discussion about AHD and cycling in and out of care as there is much literature to support this. But your results do not agree with this statement. The authors also say (line 77-79) “These findings underscore the necessity for strategies aimed at closing gaps in HIV care, promoting re-engagement interventions (48), and encouraging individuals to return before experiencing significant deterioration (49, 50).” I agree with these sentiments, but the study’s results do not show this since the majority of those with AHD after ART initiation were continuously in care and virally suppressed. While the authors have now included a sentence about patients who experience AHD despite ART and VL suppression, the rest of the paragraph needs to be reframed to be in line with the results. The authors reference the ZIMPHIA survey, but in that study only 30% of patients with advanced disease were virally suppressed. The authors have not discussed the much higher proportion of 69% that was found in this study or possible reasons for this.

o Reference 44 and 45 which are meant to support the statement do not seem to be appropriate (line 67-68): “The subset of PLWH experiencing AHD despite prolonged ART and viral load suppression has been noted in several other studies from the region (43-45).” The Ousley study is of hospitalized patients and found that between 81% and 84% of patients admitted to hospital who were ART experienced also had a detectable VL of >1000 copies/mL. This study’s findings seem to be exactly opposite to the point the authors are trying to make. The Nanzigu study (ref 45) describes immune recovery and offers no evidence about advanced disease and viral suppression.

7. PLOS authors have the option to publish the peer review history of their article (what does this mean? ). If published, this will include your full peer review and any attached files.

**Do you want your identity to be public for this peer review?** For information about this choice, including consent withdrawal, please see our Privacy Policy .

Reviewer #1: No

Reviewer #3: **Yes: ** Kerusha Govender

Reviewer #4: No

---

## [Editor Report · Acceptance letter]

PONE-D-24-02035R2

PLOS ONE

Dear Dr. Kitenge,

I'm pleased to inform you that your manuscript has been deemed suitable for publication in PLOS ONE. Congratulations! Your manuscript is now being handed over to our production team.

Kind regards,

on behalf of

Dr. Matthew J. Mimiaga

Academic Editor

PLOS ONE